# Spore-Based Probiotic *Bacillus subtilis*: Current Applications in Humans and Future Perspectives

**Natasha Williams *** and **Tiffany L. Weir ***

Department of Food Science and Human Nutrition, Colorado State University, Fort Collins, CO 80523, USA
* Correspondence: natasha.williams@colostate.edu (N.W.); tiffany.weir@colostate.edu (T.L.W.)

**Abstract:** *Bacillus subtilis* has been used for more than 50 years in many different industrial applications, including farming, precision fermentation, and probiotic supplements. It is particularly attractive as a probiotic because of its ability to form shelf-stable, acid-resistant spores that lend to diverse applications in the food system. *B. subtilis* is the most ubiquitous species of the genus and can be isolated from a broad variety of environments including animal and human gastrointestinal (GI) tracts. This is a comprehensive review of human intervention studies utilizing *B. subtilis* as a probiotic for supporting gastrointestinal health, as well as the reported impacts of B. subtilis use on the human gut microbiota and other biomarkers of health. It briefly covers the fate of ingested spores in the GI tract, summarizes the observed effects of different probiotic *B. subtilis strains*, and offers a perspective for the continued and future uses of *B. subtilis* in human applications.

**Keywords:** probiotic; *Bacillus subtilis*; gastrointestinal; microbiota

## 1. Introduction

*Bacillus subtilis* has been used for more than 50 years in many different industrial applications, including farming, precision fermentation, and probiotics [1]. *Bacillus subtilis* is particularly attractive as a probiotic because of its ability to form shelf-stable, acid-resistant spores that lend to diverse applications in the food system. *Bacillus subtilis* is the most ubiquitous species of the genus and can be isolated from a wide variety of environments: terrestrial, aquatic, food, and host-associated. It has been identified as free-living in the soil, as well as in association with a variety of plants [2,3]. Through its close association with plants, and also by releasing a multitude of airborne spores, the species easily finds its way into the gastrointestinal (GI) tract of animals. It has been isolated from the GI contents and feces of multiple species of insects [4,5], fish [6], birds [7], and mammals [8–10] in their natural environments. The wide distribution of *B. subtilis* can be attributed to three major characteristics of the species. First, in nutrient-deprived environments, it forms endospores that can remain dormant for very long periods of time before returning to an active vegetative state when conditions are favorable. Second, it has facultative respiratory pathways that allow for metabolic flexibility under both aerobic and anaerobic conditions. In an aerobic environment, *B. subtilis* uses oxygen as an electron acceptor. However, under the oxygen-deprived conditions of the GI tract of mammals, it can utilize anaerobic metabolic pathways by expressing nitrate reductase genes, which enable the utilization of nitrate instead of oxygen as a final electron acceptor [11,12]. This anaerobic pathway allows *B. subtilis* to complete its entire lifecycle in the mammalian GI tract: from the germination of the spore into its vegetative state to its proliferation and subsequent sporulation [13–16]. The third survival strategy is based on the extreme genetic polymorphisms within the species. Two phylogenetic clusters within the species were separated into subspecies *B. subtilis* subsp. *subtilis* and *B. subtilis* subsp. *spizizenii*, each containing multiple strains [17,18]. The genetic structure includes segments of the genome that are highly conserved, combined with genetically diverse elements. The most conserved

genes are responsible for sporulation, while the most divergent genes are responsible for germination, which ensures the successful survival and propagation of different strains in a wide variety of environments [19,20]. Genetic variation is augmented by horizontal gene transfer between strains via transformation, conjugation, or transduction [21]. Together, these adaptations make *B. subtilis* an incredibly competitive and versatile microorganism. In this review, we discuss *B. subtilis* in the context of its use as a probiotic for improving human gastrointestinal health.

## 2. Life Cycle of *Bacillus subtilis* in the Gastrointestinal Tract

The spores of *Bacillus* spp. are some of the hardiest biological agents on Earth. They can easily survive the harsh acidic environment of the stomach and concentration of bile salts in the duodenum, arriving unscathed in the small intestine and advancing through the rest of the GI tract where they can exert their effects. As a probiotic, the administration of spores rather than vegetative bacteria increases the shelf life and GI survival but introduces questions regarding metabolic activity and efficacy. The spores of *Bacillus* spp. germinate in nutrient-rich environments, including the environment of the stomach and small intestine, but does germination continue throughout the GI tract? How many spores from an administered dose actually germinate? What is the outgrowth of the germinated portion of the bacteria, and do they return to spore form before leaving the GI tract? Some of these very important questions were addressed in *in vitro* and/or *in vivo* studies with animal models, and, more recently, in human subjects.

A number of studies have shown that spores of different strains of *B. subtilis* germinate in the GI tract of mice [13,15,16] and chickens [22,23]. The details of these studies were previously reviewed by Bernardeau et al. [24]. The rate of germination and the ratio of spores to vegetative cells vary among studies. However, vegetative cells start appearing in the stomach of mice and persist throughout the entire GI tract, getting excreted in the feces. In addition to rodent studies, *B. subtilis* spores are also commonly administered as a probiotic to pigs [25,26], animals that are structurally and metabolically closer to humans than mice or chickens. One of the most comprehensive studies of the fate of *B. subtilis* spores in the GI tract of pigs was conducted by Lesser et al. [14] using three separate experiments. In Experiment 1, the piglets (*n* = 10) were divided into two dietary groups: control and treatment with *B. subtilis* CH201 and *B. licheniformis* CH200 as a 1:1 spore mixture added into the standard diet at $1.28 \times 10^8$ *Bacillus* spores per gram of feed. After 14 days, the contents of different parts of the GI tract were sampled. Viable spores were detected in all segments of the GI tract after feeding a diet that included *B. subtilis*. Spores were detected in the stomach (25% of the number of spores in the feed) and then increased in the small intestine and remained at that level in the caecum and colon. More spores were found in the intestines than in the stomach, indicating that some of the germinated spores may have re-sporulated after passing into the small intestine. In Experiment 2, grower-finisher pigs (*n* = 20) were divided into the same two dietary groups for 60–120 days, followed by a diet containing no supplement for an additional 7 days. After the withdrawal of the spore-containing diet, the number of spores in the fecal samples gradually declined during the 7-day period, and at that timepoint, spore counts in the feces decreased to the level of the control group. Therefore, *B. subtilis* was unable to colonize the GI tract. In Experiment 3, grower-finisher pigs (*n* = 6) orally received spores contained in dialysis tubes. Contents of different parts of the GI tract were sampled at different timepoints up to 24 h. The samples recovered from the GI tract were grouped into two categories dominated by either spores or vegetative cells, according to time. The number of spores in the dialysis tubes recovered from the large bowel was lower than in the tubes recovered from the stomach and proximal intestine. These results showed that spores of *B. subtilis* CH201 germinate in the GI tract of pigs, starting in the stomach and advancing through the rest of the GI tract. The authors concluded that about 70–90% of diet-supplemented *Bacillus* spores germinate in the proximal part of the pig GI tract, but the outgrowth of the vegetative cell population

is limited, confirming that the spores and vegetative cells of *Bacillus* transiently remain in the system but are unable to permanently colonize the GI tract.

While the previously conducted studies were informative, the fact that they were conducted in animal models limits translatability to human relevance. In human populations, most studies on the lifecycle of *B. subtilis* following spore-based probiotic administration are based on fecal sample analysis or artificial GIT models. A few studies examined the fate of *B. subtilis* spores in the simulated GIT environment [27,28], which were previously reviewed [24]. To our knowledge, there is only one study to date that has examined the lifecycle of *B. subtilis in vivo* in a functional human gut. A unique, real-time intervention trial was conducted with human subjects at the Cork Teaching Hospitals in Ireland [29] as a randomized, crossover, double-blind, placebo-controlled trial. Participants were adults (aged 18–75) with an ileostomy who were at least 3 months post-operation and otherwise healthy. Eleven participants received either *B. subtilis* DE111 probiotic spores ($5 \times 10^9$ CFU single dose) or placebo with a meal. The content of the ileal effluent was collected at baseline and every hour for 8 h post ingestion. The spores and vegetative cells of *B. subtilis* DE111 were quantified in each collected sample. Three hours following the ingestion of DE111, *B. subtilis* spores ($6.4 \times 10^4 \pm 1.3 \times 10^5$ CFU/g effluent dry weight) and vegetative cells ($4.7 \times 10^4 \pm 1.1 \times 10^5$ CFU/g effluent dry weight) appeared in the ileum effluent. Six hours after ingestion, spore concentrations increased to $9.7 \times 10^7 \pm 8.1 \times 10^7$ CFU/g and remained constant through the final time point at 8 h. Vegetative cells reached a concentration of $7.3 \times 10^7 \pm 1.4 \times 10^8$ CFU/g at 7 h following ingestion. Both the spores and vegetative cells were detected in the small intestine 3 h after ingestion of the probiotic capsule. Concentrations of vegetative cells in the ileal effluents reached a peak at 7 h after ingestion. The concentrations of spores and cells differed across timepoints for different participants. However, all participant samples had spores present 5 h after ingestion and had vegetative cells present at some time throughout the session. Germinated spores of *B. subtilis* DE111 were detected in the lower ileum at different concentrations during the 3 to 8 h post-ingestion period. The study showed that orally ingested *B. subtilis* DE111 spores remain viable during their transit through the upper GI tract and are able to germinate in the small intestine of humans. Based on the results of these studies, it can be concluded that both spores and vegetative cells of orally administered *B. subtilis* are present in different parts of the GI tract at different ratios.

Since *Bacillus* spores are metabolically inert entities, it has been hypothesized that the main way they exert their action is through a "passive effect", eliciting a response from cells of the GI tract and/or intestinal microbiota upon detection. In contrast, the vegetative cells are metabolically active and can influence the GI environment in many different ways, exerting an "active" effect. The other way to classify the mode of action of both the spores and vegetative cells is to consider their effects as direct or indirect, where they are either directly affecting the cells of the GI tract or indirectly affecting the GI tract's environment through a modification of the commensal microbiota. Therefore, there are four possible modes of action for spore-based probiotics: passive direct, passive indirect, active direct, and active indirect (Figure 1). Which of these modes contribute to the probiotic effects of ingested spores and to what extent is an intriguing question that is currently being investigated through multiple human studies.

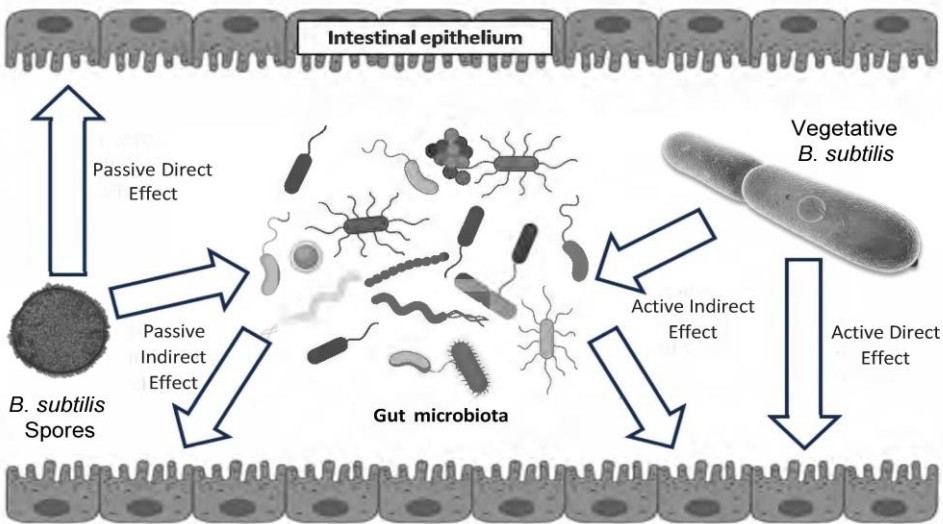

**Figure 1.** Possible mechanistic targets of spore or vegetative cells of probiotic *Bacillus subtilis* in the human gastrointestinal tract.

### 3. Effects of Different Strains of Probiotic *Bacillus subtilis* in Human Applications

The multifaceted effects of probiotic B. *subtilis* have been extensively studied in animal models. Recent research has evaluated the effects of different strains of *B. subtilis* on health maintenance and disease resistance in pigs [30,31], dogs [32–35], and chickens [36–39] in terms of the probiotic potential to enhance health in pets and health/productivity in farm animals. However, this review primarily focuses on recent research utilizing various strains of *B. subtilis* in human populations.

Clinical investigations of the probiotic effects of *B. subtilis* have been ongoing for the last several decades. As previously reviewed [40,41], earlier studies focused on the role of *B. subtilis* in individuals with intestinal infections and defined disease states, including but not limited to antibiotic-associated diarrhea, ulcerative colitis, acute enteric infections, irritable bowel syndrome, infectious pathologies of different origins, therapy for *Helicobacter pylori* eradication, and dysbacteriosis in neonates and children. With the emergence of Next Gen sequencing techniques, more recent studies have examined the impact of *B. subtilis* administration on intestinal microbiota interactions. The subject population largely changed from ill patients in hospital settings towards generally healthy individuals who could potentially benefit from probiotic supplementation in their daily lives.

While, *B. subtilis* has been investigated as a component of multi-species probiotics in several recent clinical trials [42–45], these studies are not included in the current review. Instead, we comprehensively summarize studies examining the effects of different strains of *B. subtilis* administered as a single-organism probiotic (Figure 2). Table 1 summarizes findings of research conducted between 2015 and 2023 using single strains of *B. subtilis* administered orally in an encapsulated form to examine effects on gastrointestinal health in humans, alteration of the GI microbiota, and other parameters of health and well-being. These comprehensive studies examined multiple subjective and objective outcomes, including but not limited to GI health and quality of life questionnaires, physical activity and diet records, daily bowel movement charts, analysis of body fluids (blood, urine, saliva), and intestinal microbiota analysis based on 16S rRNA gene sequencing. The subjects also represent diverse human populations, including healthy adults, children in daycare, college athletes, post-menopausal women, the elderly, and people with different degrees of mild-to-moderate gastrointestinal distress.

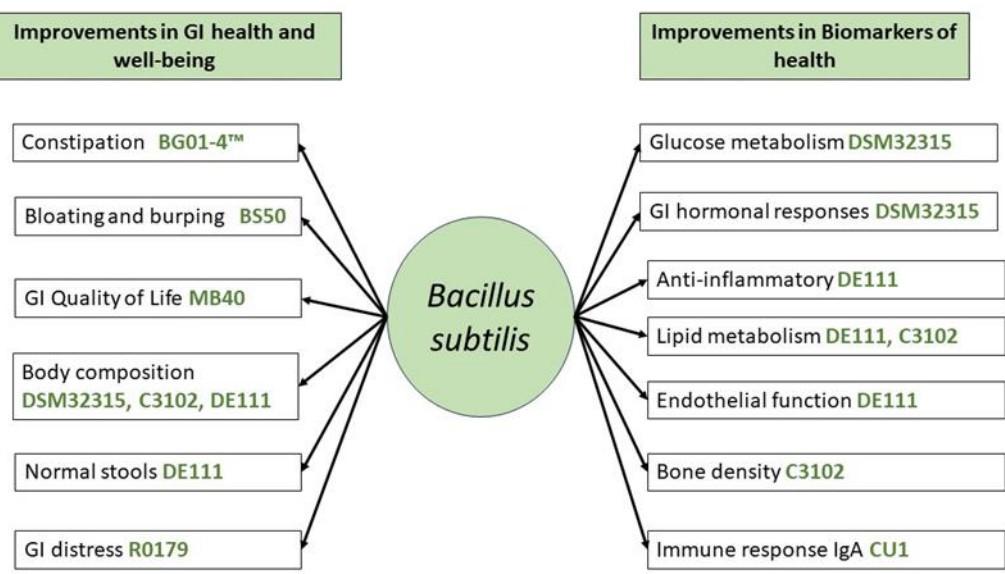

**Figure 2.** Strain-specific effects of probiotic *Bacillus subtilis* in human applications.

### 3.1. Bacillus subtilis R0179

One of the earlier studies that included microbiota analysis [46] aimed to establish the oral dose–response tolerance and gastrointestinal viability of *B. subtilis* R0179 in human subjects (Table 1). The outcomes included daily questionnaire analysis (GI distress, cephalic, epidermal, ear-nose-throat, behavioral and emetic syndrome scores), survival of probiotic after gastrointestinal transit, and microbiota composition analysis. The study concluded that probiotic *B. subtilis* R0179 was well tolerated at all tested doses ($0.1 \times 10^9$ to $10 \times 10^9$ CFU), did not persist in the human GI tract, and did not significantly affect local microbiota at the phylum level, though some changes occurred at lower taxonomic scales (i.e., increase in operational taxonomic units matching most closely to *Ruminococcus*). Interestingly, while *B. subtilis* R0179 did increase the prevalence of some taxa, it was more strongly associated with a decrease in other taxa. The authors speculate that supplementation might elicit some competitive inhibitory effects on the growth of undesired opportunistic pathogens.

### 3.2. Bacillus subtilis DE111

A series of studies have examined the effects of probiotic strain *B. subtilis* DE111 on different subsets of humans. The most comprehensive set of outcomes were evaluated in a pilot study examining *B. subtilis* DE111 effects in a healthy adult population [47]. The major findings are presented in Table 1; however, we highlight several findings that could be explored in future research. Concerning gastrointestinal health, participants consuming DE111 showed a trending reduction in symptom severity scores related to perceptions of gastric function, relative to their baseline ($p = 0.056$; CI = $-3.49$, $0.04$). Gastrointestinal inflammation, small intestine and pancreas, and colon pain scores did not change over time with the probiotic intervention or placebo. A microbiota analysis did not detect any significant differences in alpha diversity or in Bray–Curtis distances between intervention groups. However, the most interesting findings were related to systemic immune responses. Immune cell populations in peripheral blood mononuclear cells (PBMCs) were quantified employing two approaches. First, collected PBMCs were assessed through flow cytometry to determine basal immune cell populations both pre- and post intervention. DE111 intervention significantly decreased CD3+ T cells and CD25+FoxP3+ T regulatory cells when comparing baseline to post-intervention. There was also an observed trend for DE111 intervention to decrease CD4+ T helper cells compared to placebo. There was no effect on myeloid cells or B cells. In a second approach, collected and cultured PBMCs were stimulated with bacterial lipopolysaccharide (LPS). The immune cell response was calculated as a ratio of LPS-stimulated cell counts/basal cell counts. Significant increases

were observed for CD25+ and CD25+FoxP3+ T regulatory cells, and CD4+CD8+ double-positive T cells, while CD8+ cytotoxic T cells were reduced after DE111 intervention compared to the placebo group, suggesting immune suppression under basal conditions, but increased ability to respond to inflammatory stimuli. Other cell types were unchanged, as were circulating inflammatory markers and markers of intestinal permeability. These data suggest a possible effect of probiotic *B. subtilis* DE111 on the immune system of the human population, which warrants further exploration.

A microbiota composition analysis following 8 weeks of *B. subtilis* DE111 supplementation was performed on stool samples of healthy children attending daycare (Table 1) [48]. At the phylum level, the intake of *B. subtilis* DE111 significantly increased microbial community diversity (Shannon and Simpson indices) without globally shifting the equilibrium of the microbiome. However, there were changes in the differential abundance of some taxa at the genus level that could be considered beneficial. Members of *Alistipes*, *Bacteroides*, *Parabacteroides*, *Odoribacter*, and Rikenellaceae increased in the probiotic group. According to the authors' statements, representatives of these taxa are implicated in immune regulation and reduction of inflammation, while the decreased taxa that included *Eisenbergiella*, *Lactobacillales*, and *Streptococcaceae* may be considered pro-inflammatory. Thus, the increased diversity and specific taxa changes suggest a shift towards a healthier microbiota composition following probiotic supplementation. A decrease in the *Bacillota/Bacteriodota* (formerly referred to as *Firmicutes/Bacteroidetes*) ratio following probiotic supplementation was also observed. Gastrointestinal health, or any other characteristics, were not evaluated in this study.

The effects of *B. subtilis* DE111 on stool profiles were investigated by Cuentas et al. [49] in healthy adults suffering from occasional constipation and/or diarrhea (Table 1). The study evaluated GI health using the Bristol stool chart and digestive health questionnaires. Blood samples were collected at three timepoints during the study and analyzed for C-reactive protein, lipid profiles, and comprehensive metabolic panels. The authors reported improvements in stool type (normal vs. abnormal) in the probiotic group. No other effects of supplementation on GI health were reported. All blood markers stayed within normal refence ranges for both probiotic and placebo groups, and no changes were recorded for biomarkers throughout the study. According to the authors' conclusions, *B. subtilis* DE111 can help to maintain gastrointestinal health by improving occasional constipation and/or diarrhea.

An exploratory study on the use of *B. subtilis* DE111 supplementation in college athletes and physically active adults was completed at the Human Performance Laboratory at Lipscomb University. A study by Townsend et al. [50] was the first to examine the potential benefits of probiotic *B. subtilis* DE111 supplementation in male college athletes during offseason training (Table 1). Body composition was evaluated as an indicator of athletic status pre- and post-training season. Dynamic strength, ten-yard sprint, pro-agility test, and standing long jump were primary outcomes for testing athletic performance. Biochemical analyses included measurements of salivary immunoglobulins SIgA and SIgM, which were used as indicators of mucosal immunity. Blood samples were analyzed for the following markers: TNF-$\alpha$, IL-10, zonulin, testosterone, and cortisol. Though, *B. subtilis* DE111 supplementation was well tolerated by athletes, it did not affect body composition, performance, hormonal concentrations, and gut permeability, but it did result in lowering blood concentrations of TNF-$\alpha$. According to the authors' conclusions, attenuating circulating TNF-$\alpha$ concentrations in college athletes following offseason training may be beneficial; however, the relevance of this effect on overall health is still unexplored. A follow-up study [51] examined the effects of the similar *B. subtilis* DE111 supplementation on female college athletes during their offseason resistance training. The measured outcomes were limited to body composition and resistance performance. No analysis of biological samples was conducted in this study. Probiotic supplementation did not affect athletic performance, but it improved body composition (Table 1). The next study conducted in the same Human Performance Lab [52] determined if probiotic *B. subtilis* DE111 supplementation influenced

plasma amino acid (AA) response to acute whey protein ingestion in physically active adults. Fasting blood samples were collected at baseline and post-treatment visits from time zero at 15 min intervals for 2 h after ingestion of 25 g of whey protein dissolved in 400 mL of water. The following amino acids were quantified in blood plasma: leucine, branched-chain AA, essential AA, and total AA. The study did not find any significant differences between treatment and placebo groups and concluded that DE111 supplementation does not affect protein utilization in exercising adults.

Trotter et al. [53] explored the potential health effects of probiotics administered alone or concurrently with bacteriophages. One of the probiotics tested was *B. subtilis* DE111, which was administered as a single strain as one arm of this study. A pilot exploration aimed to determine whether the four-week consumption of (1) maltodextrin placebo; (2) *Bifidobacterium lactis* alone, or (3) *Bifidobacterium lactis* in combination with a cocktail of *E. coli*-targeting bacteriophages; and (4) *Bacillus subtilis* DE111 altered risk factors for CVD. The primary outcome measures included blood pressure, endothelial function, and plasma lipid profiles. Researchers hypothesized that probiotic consumption would improve one or more measures of cardiovascular function in a healthy adult population, and that simultaneous supplementation with *E. coli*-targeting bacteriophages might further enhance these beneficial cardiovascular effects [53]. Interestingly, the authors did not find any significant changes in measured CVD parameters among individuals consuming *Bifidobacterium lactis* with or without bacteriophages. However, supplementation with *B. subtilis* DE111 resulted in a significant reduction in total cholesterol and non-high-density lipoprotein cholesterol relative to baseline measures. There were also modest, but clinically relevant, improvements in endothelial function and low-density lipoprotein cholesterol following the consumption of *B. subtilis* supplements. The authors concluded that *B. subtilis* DE111 supplementation may be beneficial for improving risk factors associated with CVD (Table 1).

### 3.3. Bacillus subtilis C-3102

Three human studies have looked at the various health impacts of consuming *B. subtilis* C-3102 in Japanese cohorts. Takimoto et al. investigated the effect of this probiotic on bone health in post-menopausal women [54] (Table 1). The outcomes included bone mineral density (BMD) measured at the lumbar spine and hip using dual-energy X-ray absorptiometry and markers of bone turnover. Markers of bone resorption included urinary type I collagen cross-linked N-telopeptide (uNTx) and serum tartrateresistant acid phosphatase isoform 5b (TRACP-5b). The markers of bone formation included serum bone alkaline phosphatase (BAP) and intact parathyroid hormone (iPTH). Also, microbiota composition analysis was performed on fecal samples. The measurements and sample collection were performed at baseline and at 12-week and 24-week treatment periods. After 24 weeks of probiotic supplementation, total hip BMD significantly increased; however, there was no significant difference between the probiotic and placebo groups for lumbar spine BMD. Both markers of bone resorption were decreased in the probiotic group after 12 weeks of supplementation; however, there was no significant difference between the placebo and C-3102 groups in these two bone resorption markers at 24 weeks of treatment. No significant changes were recorded for the bone formation markers at either timepoints. Gut microbiota analysis showed a decrease in Chao1 and Shannon indices of alpha-diversity after 24 weeks of probiotic treatment. The differential abundance analysis at the genus level showed a relative increase in *Bifidobacterium* in the C-3102 group at 12 weeks of treatment when compared with the baseline, and genus *Fusobacterium* significantly decreased in the C-3102 group at 12 and 24 weeks of treatment when compared with the baseline. The authors concluded that the results were suggestive of the positive effects of probiotic strain *B. subtilis* C-3102 on bone health in post-menopausal women. Also, they are suggestive that *B. subtilis* C-3102 modulates host-gut microbiota; however, specific microbiota modifications did not significantly correlate with BMD or bone turnover markers.

The second Japanese study evaluated the possible preventive effects of the ingestion of *B. subtilis* C-3102 on chronic diarrhea in healthy volunteers with loose stools [55]. The study utilized gastrointestinal health and quality of life questionnaires, Bristol stool chart, determination of fecal water content, and microbiota analysis. Several parameters were significantly improved via C-3102 treatment (Table 1). Gut microbiota analysis revealed no changes in alpha-diversity after 8 weeks of C-3102 ingestion. However, the relative abundance of two genera in the gut microbiota (*Lachnospira* and *Actinomyces*) was significantly changed: *Lachnospira* increased in relative abundance, and *Actinomyces* decreased post treatment. The authors concluded that improvement in bowel habits may be related to the modifications in gut microbiota in response to C-3102 ingestion.

Finally, the third study evaluated the safety of excessive consumption of *B. subtilis* C-3102 in healthy volunteers [56]. The outcomes were based on anthropometric parameters, blood hematological tests (including complete white blood cell count), very extensive blood biochemical analyses, urinalyses, and measurements of bone mineral density. Subjects also completed a medical questionnaire to determine their health status at three assessment points (baseline, 2-week, and 4-week). The major findings are presented in Table 1. In addition, some differences were observed between males and females: the cholinesterase levels were significantly higher in female subjects in the C-3102 group than in the placebo group after 2 weeks of probiotic intake, while there were no changes in the blood parameters in males. No changes were reported for medical questionnaire reports, urinalysis, and BMD. Moreover, all the reported changes remained within clinical reference ranges and did not indicate any medical conditions or complications. Therefore, the consumption of excessive amounts of probiotic *B. subtilis* C-3102 was determined to be safe by the investigators.

### 3.4. Bacillus subtilis BS50

A unique strain *B. subtilis* BS50 that was isolated from soil and showed promise as a probiotic was evaluated by Brutscher et al. [57] in preclinical trials. Before any clinical trial of a new strain can occur, the safety profile should be evaluated in preclinical testing. This study screened the genome for genes encoding virulence factors, *Bacillus* toxins, and antibiotic resistance. Cultured human intestinal epithelial cells (Caco-2) were used to perform viability and permeability assays. Several gene clusters were identified that are involved in the biosynthesis of secondary antimicrobial metabolites that do not present any harm to the intestinal cells. The study concluded that BS50 was unlikely to negatively affect human enterocytes or disrupt gut barrier integrity. A follow-up clinical trial investigated the safety and efficacy of the daily supplementation of *B. subtilis* BS50 for 6 weeks in healthy adults who had mild gastrointestinal distress before the start of supplementation [58]. Intestinal distress was defined as having a combined score of 3 or more for abdominal bloating, burping, and flatulence by assessment using the Gastrointestinal Tolerance Questionnaire (GITQ) (Table 1). Besides GI symptoms assessment using the GITQ, the other outcomes included a bowel habit diary, sleep quality and respiratory infection questionnaire, and blood sample analysis. Fasting blood samples were analyzed for markers of intestinal permeability (zonulin, occludin, and LBP), inflammatory markers (CRP, IL-8, IL-6, IL-10, IFN-$\gamma$, and TNF-$\alpha$), and lipid profiles (TG, total-C, HDLC, LDL-C). Six weeks of *B. subtilis* BS50 supplementation significantly improved GI symptoms (bloating, burping, and flatulence) as assessed using the GITQ (Table 1). Bowel habits did not significantly change with the intervention. The number of bowel movements slightly increased during the weeks of supplementation but was not significantly different from placebo. No changes were recorded for the symptoms of discomfort during bowel movement, straining, or feeling of incomplete evacuation. There was no effect of supplementation on the quality of sleep and respiratory infections. The markers of intestinal permeability, lipid profiles, and most inflammatory markers were not affected by the treatment. There was a slight increase in circulating anti-inflammatory cytokine, IL-10, in the *B. subtilis* BS50 group vs. placebo, but it failed to achieve statistical significance. The authors concluded that daily oral supplementation with probiotic *B. subtilis* BS50 was safe and well tolerated and im-

proved the composite score for bloating, burping, and flatulence, compared to placebo. Supplementation may be recommended to alleviate gas-related gastrointestinal symptoms in a generally healthy population.

### 3.5. Bacillus subtilis MB4

A subset of gastrointestinal distress symptoms, i.e., bloating, abdominal discomfort, and gas, were the primary outcomes in Penet et al.'s study examining the effects of *B. subtilis* MB40 supplementation (Table 1) [59]. During the 4 weeks of treatment, participants completed three questionnaires daily: a modified Abdominal Discomfort, Gas, and Bloating (mADGB) questionnaire, a modified Gastrointestinal Symptoms Rating Scale (mGSRS), and a Bowel Habits Diary (BHD). For the quality-of-life and general health assessment, a modified RAND SF-36 questionnaire was filled out at baseline and weeks 2 and 4. Blood samples were analyzed for hematologic and chemical parameters to evaluate the safety of probiotic administration. All blood parameters were within the normal clinical ranges before and after the treatment. At the end of 4 weeks, the study did not find any significant differences between MB40 and placebo groups in the average weekly number of days with bloating, bloating intensity, or abdominal discomfort and gas. However, the male sub-group in MB40 showed clinically relevant improvements in some of those parameters and also reported improvements in some quality-of-life characteristics and the general health score (Table 1). The authors concluded that *B. subtilis* MB40 supplementation was safe and well-tolerated but did not significantly improve major outcome parameters in the MB40 group versus the placebo.

### 3.6. Bacillus subtilis CU1

Lefevre et al. investigated the effect of probiotic strain *B. subtilis* CU1 intake on resistance to common infectious diseases (CIDs) in healthy, free-living seniors [60]. The primary outcome was the mean cumulative number of days with CID in participants. The secondary outcomes determined the effect of *B. subtilis* CU1 intake on the stimulation of the mucosal and systemic immune response by measuring intestinal and salivary sIgA levels and serum cytokine levels in a subset of 44 subjects out of 100 subjects completing the trial (Table 1). The primary outcomes did not show any statistically significant difference between the probiotic and the placebo groups in mean duration, intensity, and frequency of CID during the observation period. In the subset of 44 individuals, the frequency of respiratory infections was significantly lower in the probiotic group compared to the placebo group. The significant increases were also reflected in intestinal and salivary SIgA levels in the probiotic group compared to the placebo group. Also, IFN-gamma concentrations significantly increased in the probiotic group after 10 days of probiotic consumption, while no change was observed for the placebo. The results of the study indicate that *B. subtilis* CU1 can modulate the immune response in the elderly population; however, no definite conclusion can be made about the effect of *B. subtilis* CU1 supplementation on CID.

### 3.7. Bacillus subtilis-Containing Synbiotic Products

Synbiotics are a combination of a probiotic with a prebiotic which can act either synergistically, where the prebiotic specifically supports the growth and survival of the probiotic strain, or in a complementary manner, where each component exerts independent beneficial effects on the GI tract. A novel complementary synbiotic formulation of the strain *B. subtilis* DSM32315 was tested on human subjects in a single-arm study (Table 1) [61]. The synbiotic formulation was developed in the laboratory in Germany and included *B. subtilis* DSM32315 and L-Alanyl-L-Glutamine as main ingredients, plus plant extracts, minerals, and vitamins (SAMANA® Force, Evonik, Darmstadt, Germany). By using a probiotic in combination with an amino acid, the researchers were targeting butyrate-producing commensal microbes that are able to process peptides and amino acids as substrates in the pyruvate/acetyl-CoA pathway. The study was based on the presumption that probiotic *B. subtilis* can modulate the human colonic microbiota towards an increase

in pro-butyrogenic species that can selectively use a stable, non-fiber substrate L-alanyl-L-glutamine for butyrate production. Healthy males with "health unconscious eating patterns" consumed the formulation daily for 4 weeks. The blood and fecal samples were collected at baseline, 2-week, and 4-week timepoints. The microbiota analysis and quantification of short-chain fatty acids (SCFAs) were performed on the fecal samples. Blood was analyzed for lipid profiles, fasting glucose, and the hormones PYY (Peptide YY) and GLP-1 (Glucagon-like Peptide 1). The focus of the study was to determine the effects of the synbiotic formulation on the levels of SCFAs in feces. There was a significant increase in butyrate levels (21%) after 2 weeks in post-treatment samples, with no significant changes in the levels of propionate and acetate. A microbiota alpha-diversity analysis revealed no change in richness indices but did show a decrease in the Pielou's Index of Evenness in post-treatment samples, which was interpreted by the researchers as a shift in the proportion of different taxa within samples. Differential analyses revealed that the abundance of 20 species significantly changed in post-treatment compared to baseline samples; the most noticeable were increased levels of *Faecalibacterium prausnitzii*. Blood biomarker quantification indicated a reduction in both measured hormones and changes in lipid profiles (Table 1). In addition, fasting glucose concentrations showed a trend of reduction in post-treatment compared to pre-treatment samples; however, the change did not reach statistical significance. The results of the study indicate that the tested synbiotic formulation may be an effective way to stimulate intestinal butyrate production, with additional influence on lipid and glucose profiles.

A follow-up exploratory human study used the same synbiotic supplementation [11]. A cohort of men and women consumed two capsules of synbiotic formulation (SAMANA® Force, Evonik, Darmstadt, Germany) daily for 4 weeks. Blood and fecal samples were collected at baseline and post-treatment. Participants filled out several questionnaires throughout the intervention, which captured the supplement's effects on digestive parameters, frequency and consistency of bowel movements, feelings of hunger and satiety, well-being, and physical performance. The main results for the total cohort are presented in Table 1. Briefly, investigators showed that the supplement use was associated with significant decreases in fasting blood glucose and glycated hemoglobin, HbA1c, a stable marker of blood glucose and indicator of the diabetic status of an individual. However, based on the baseline fasting glucose levels and HbA1c values, participants were divided into two subgroups: prediabetic ($n = 62$) and non-prediabetic (healthy). The subgroup analysis revealed that improvements in fasting glucose, HbA1c values, and glycemic response were driven by the prediabetic subgroup. Significant weight loss was recorded for both subgroups, and the overall average was $1.07 \pm 2.30$ kg; specifically, the differences before and after the study were $1.47 \pm 2.82$ kg and $0.87 \pm 1.97$ kg, respectively, for prediabetic and healthy participants. Questionnaires did not record major changes during supplementation, except for the feeling of hunger, which was reduced significantly towards the end of the study. Microbiome analysis also reflected different reactions between subgroups. Shannon's index of alpha-diversity was increased significantly post-treatment in the healthy subgroup but did not change in the prediabetic subgroup. In the total population, *Bacteroidota* (formerly *Bacteroidetes*) significantly increased throughout the observation, and the abundance of *Bacillota* (formerly *Firmicutes)* decreased, and these changes were largely driven by the prediabetic population. In healthy participants, no significant differences were observed at the phylum levels between the start and end of the observation. The authors suggested that the tested supplement may be recommended for the management of hyperglycemia and metabolic syndrome.

### 3.8. Bacillus subtilis as a Postbiotic

A new proprietary strain of inactivated *B. subtilis* BG01-4™ high in branched-chain fatty acids (BCFA) was used to treat participants with self-reported diagnoses of functional gastrointestinal disorders (FGIDs) [62]. The effects were evaluated based on the Gastrointestinal Symptom Rating Scale (GSRS) questionnaire filled out at baseline, 2-week, and

4-week timepoints. Three primary outcomes included Total GSRS score, GSRS-constipation, and GSRS-diarrhea, while secondary outcomes were indigestion, dyspepsia, and abdominal pain syndrome. Based on the results of the study (Table 1), the authors concluded that postbiotic *B. subtilis* BG01-4™ can improve specific symptoms of constipation and related GI dysfunction in people with a FGID.

A summary of the results of the above-presented studies in terms of *B. subtilis*'s possible positive effects on gastrointestinal health shows improvements in constipation, indigestion, and dyspepsia [62]; bloating and burping [58]; quality of life and physical functioning related to gastrointestinal conditions [59]; stool frequency and quality [55]; abdominal sound symptoms [55]; and the proportion of normal stools [49]. Many studies reported shifts in microbiota composition following *B. subtilis* administration [46,48,54,55,61]; however, these modifications warrant further analyses in terms of their overall effects on gastrointestinal health.

Table 1. Overview of probiotic *B. subtilis* applications in human studies.

| Reference | Study Design Subjects/Models | Probiotic Dose/Duration | Results |
|---|---|---|---|
| Patch et al. 2023 [62] | Randomized, double-blind, placebo-controlled, parallel-arm trial. Healthy adults (aged 18–75), *n* = 67, with self-reported diagnosis of functional gastrointestinal disorders (FGID). | B. subtilis BG01-4™ $5 \times 10^9$ CFU Daily dose for 4 weeks | Constipation in the probiotic group was significantly improved compared to placebo (33% vs.15%, respectively). Clusters for constipation (18% improvement), indigestion (11%), and dyspepsia (10%) were significantly improved in the probiotic group compared to the placebo. |
| Garvey et al. 2022 [58] | Randomized, double-blind, placebo-controlled, parallel-arm clinical trial. Healthy adults (aged 30–65), *n* = 76, with at least minimal complaints of abdominal bloating, burping, or flatulence. | B. subtilis BS50 $2 \times 10^9$ CFU 1 capsule/day for 6 weeks | Improvement of 2 or more points in the 7-day, 3-item composite score according to GITQ (composite score for flatulence, bloating, and burping) between baseline and week 6 (47.4% vs. 22.2%). Compared to placebo, the proportion of participants with an improvement of 1 or more points in GITQ for burping (44.7% vs. 22.2%) and bloating (31.6% vs. 13.9%). There were no significant differences between groups for flatulence (47.4% vs. 44.4%). No change in bowel habits, sleep quality, respiratory infections, and blood markers for intestinal permeability, inflammation, and lipid profile. |
| Kordowski et al. 2022 [11] | Open-label, single-arm real-life exploratory study. Healthy adults, *n* = 192. | B. subtilis DSM32315 $2 \times 10^9$ CFU (+290 mg L-Alanyl-L-Glutamine) 2 capsules/day for 4 weeks | Fasting glucose significantly decreased from pre- to post-treatment (96.92 ± 8.29 mg/dL vs. 94.58 ± 9.27 mg/dL, respectively). HbA1c significantly decreased from pre- to post-treatment (5.72% ± 0.27 vs. 5.65% ± 0.30). Postprandial glycemic response improved. Body weight (and BMI) significantly decreased. Relative abundance of Bacteroidetes significantly increased and Firmicutes decreased at post-treatment. |
| Dieck et al. 2021 [61] | Open-label, single-arm pilot study. Healthy men (aged 18–40), *n* = 18. | B. subtilis DSM32315 $2 \times 10^9$ CFU (+290 mg L-Alanyl-L-Glutamine) Daily dose for 4 weeks | DSM32315 increased levels of butyrate and butyrate-producing taxa in gut microbiota. Plasma LDL-, total cholesterol, and LDL/HDL cholesterol ratio significantly decreased. Fasting levels of PYY (Peptide YY) and GLP-1 (Glucagon-like Peptide 1) significantly decreased. |
| Freedman et al. 2021 [47] | Randomized, double-blind, placebo-controlled, parallel-arm clinical trial. Healthy adults (aged 20–65), *n* = 44. | B. subtilis DE111 $1 \times 10^9$ CFU 1 capsule/day for 4 weeks | Increase in anti-inflammatory immune cell populations in response to ex vivo LPS stimulation of PBMCs in the DE111 group. Overall perceived gastrointestinal health, microbiota, and circulating and fecal markers of inflammation (Il-6, sIgA) and gut barrier function (plasma zonulin) were largely unaffected by DE111 intervention. |

**Table 1.** *Cont.*

| Reference | Study Design Subjects/Models | Probiotic Dose/Duration | Results |
|---|---|---|---|
| Penet et at. 2021 [59] | Randomized, double-blind, placebo-controlled, parallel-arm clinical trial. Healthy adults (aged 18–75), $n = 100$, with self-reported symptoms of bloating, abdominal discomfort, and gas. | B. subtilis MB40 $5 \times 10^9$ CFU 1 capsule/day for 4 weeks | No significant differences in bloating intensity, number of days with and duration of bloating, abdominal discomfort, and gas between MB40 and placebo groups. Physical limitation, vitality, and social functioning were significantly improved from baseline to week 4 in the MB40 group. At 2 weeks, physical functioning significantly improved in the MB40 group versus placebo. Clinical, but not statistically significant (10%), reductions in bloating intensity, number of days with abdominal discomfort, gas, bloating, and duration of gas, and 10% improvement in general health score in male sub-group receiving MB40 compared to placebo. |
| Trotter et al. 2020 [53] | Randomized, double-blind, placebo-controlled, parallel-arm clinical trial. Healthy adults (aged 18–65), $n = 88$. | B. subtilis DE111 $1 \times 10^9$ CFU 1 capsule/day for 4 weeks | Significant reduction in total cholesterol and non-high-density lipoprotein cholesterol in DE111 group. Improvements in endothelial function and in low-density lipoprotein cholesterol. |
| Paytuvi-Gallart et al. 2020 [48] | Randomized, double-blind, placebo-controlled, parallel arm study. Healthy children (aged 2–6), $n = 101$, attending daycare. | B. subtilis DE111 $1 \times 10^9$ CFU 1 capsule/day for 8 weeks | Microbiome composition analysis: alpha diversity increased in probiotic group; no significant changes in the overall microbiome equilibrium; six taxa (at the genus level) significantly increased after probiotic intake, and three taxa significantly decreased. |
| Hatanaka et al. 2020 [56] | Randomized, double-blind, placebo-controlled, parallel-arm study. Healthy adults, $n = 44$. | B. subtilis C-3102 $4.8 \times 10^{10}$ CFU Daily dose for 4 weeks | Body fat percentage was significantly lower in the C-3102 group than in the placebo group at 2 weeks after probiotic. Mean corpuscular hemoglobin level was significantly higher, and cholinesterase, total cholesterol, and triglyceride levels were significantly lower 2 weeks after intake in the C-3102 group than in the placebo group. Direct bilirubin was significantly higher and total cholesterol significantly lower 4 weeks after intake in the C-3102 group than in the placebo group. No significant changes in other measured parameters. |
| Townsend et al. 2020 [52] | Randomized, double-blind, placebo-controlled, parallel-arm study. Recreationally active adults, $n = 22$. | B. subtilis DE111 $1 \times 10^9$ CFU 1 capsule/day for 28 days | Supplementation with DE111 does not affect plasma amino acid response following acute whey protein ingestion. |
| Toohey et al. 2020 [51] | Randomized, double-blind, placebo-controlled, parallel-arm study. Division I college female athletes, $n = 23$. | B. subtilis DE111 $1 \times 10^9$ CFU 1 capsule/day for 10 weeks | Significant reduction in body fat % in DE111 supplementation group ($-2.05 \pm 1.38\%$) compared with placebo ($0.2 \pm 1.6\%$). No other differences between probiotic and placebo groups were observed. |
| Townsend et al. 2018 [50] | Randomized, double-blind, placebo-controlled, parallel-arm study. Division I college male athletes, $n = 25$. | B. subtilis DE111 $1 \times 10^9$ CFU 1 capsule/day for 12 weeks | TNF-$\alpha$ concentrations were significantly lower after DE111 compared to placebo. No significant group differences in any other measured biochemical markers. No effect on body composition, performance, hormonal status, or gut permeability. |

**Table 1.** *Cont.*

| Reference | Study Design Subjects/Models | Probiotic Dose/Duration | Results |
|---|---|---|---|
| Takimoto et al. 2018 [54] | Randomized, double-blind, placebo-controlled, parallel-arm study. Healthy postmenopausal Japanese women (aged 50–69), $n = 76$. | B. subtilis C-3102 $3.4 \times 10^9$ CFU Daily dose for 24 weeks | Significant increase in total hip BMD in probiotic group (placebo = 0.83 ± 0.63%, C-3102 = 2.53 ± 0.52%). Significantly lower uNTx probiotic vs. placebo group at 12 weeks of treatment. A trend of a decrease in the bone resorption marker TRACP-5b when compared with the placebo group at 12 weeks of treatment. No change in markers of bone formation, BAP and iPTH. Changes in microbiota composition after C-3102 supplementation. |
| Hatanaka et al. 2018 [55] | Randomized, double-blind, placebo-controlled, parallel-arm study. Healthy adults (aged 20–79), $n = 82$, with loose stools. | B. subtilis C-3102 $2.2 \times 10^9$ CFU Daily dose for 8 weeks | Stool frequency per day significantly decreased after C-3102 treatment. Stool quality (measured by BBC scores) significantly improved. Abdominal sound symptoms (reported by GSRS) significantly decreased. Change in microbiota composition following C-3102 treatment. |
| Cuentas et al. 2017 [49] | Randomized, double-blind, placebo-controlled, parallel-arm clinical trial. Healthy adults (aged 18–65), $n = 50$, with occasional constipation and/or diarrhea. | B. subtilis DE111 $1 \times 10^9$ CFU 1 capsule/day for 90 days | By day 90, the proportion of normal stools (43.1%) to non-normal stools (6.13%) in the DE111 group differed significantly from placebo group (evaluated by BSC). The proportion of normal stools increased from week 1 to the last week in DE111 group (37.36% to 43.1%) vs. no change in placebo (33.77% to 35.43%). |
| Lefevre et al. 2015 [60] | Randomized, double-blind, placebo-controlled, parallel-arm study. Healthy elderly (aged 60–74), $n = 100$. | B. subtilis CU1 $2 \times 10^9$ CFU 1 capsule/day for 10 days, intermittent with 18 days, no ingestion, for 4 months | No significant decrease in mean number of days of reported for CID symptoms over the 4 months of study. B. subtilis CU1 significantly increased fecal and salivary secretory IgA concentrations compared to the placebo. No statistically significant differences in the plasma concentrations of cytokines (IL-1beta, IL-4, IL-6, IL-8, IL-10, IL-12p70, IgA, and TNF-alpha) between the probiotic and the placebo groups from pre- to post-supplementation. |
| Hanifi et al. 2015 [46] | Randomized, double-blind, placebo-controlled, parallel-arm clinical trial. Healthy adults, $n = 81$. | B. subtilis R0179 $0.1 \times 10^9$, $1.0 \times 10^9$, or $10 \times 10^9$ CFU 1 capsule/day for 4 weeks | The scores of the GI distress syndrome between placebo, 0.1, 1.0, and $10 \times 10^9$ CFU groups were equivalent. The $0.1 \times 10^9$ CFU (0.3 ± 0.1) group was not equivalent to the $1 \times 10^9$ (0.6 ± 0.1). The abdominal pain, reflux, diarrhea, indigestion, and constipation syndrome were equivalent across all periods by treatment comparisons. Microbiota composition was affected by probiotic treatment. |

Notations: CFU—Colony-forming units. GITQ—Gastrointestinal Tolerance Questionnaire. HbA1c—glycated hemoglobin (a stable indicator of glucose status and indicator for diabetes). Caco-2—cultured human intestinal epithelial cells. LPS—lipopolysaccharides. PBMC—peripheral blood mononuclear cells. BMD—bone mineral density. uNTx—urinary type I collagen cross-linked N-telopeptide. TRACP-5b—tartrateresistant acid phosphatase isoform 5b. BAP—bone alkaline phosphatase. iPTH—intact parathyroid hormone. BSC—Bristol stool chart. GSRS—gastrointestinal symptom rating scale. CID—common infectious disease.

## 4. Future Perspectives for Probiotic *Bacillus subtilis* in Human Applications

A comparative review of the human clinical trials reveals noticeable inconsistencies in the results of these studies. This effect is observable even looking at the same strain and similar outcomes; however, when comparing different strains, study outcomes can be even more divergent. The majority of studies listed in this review used probiotic *B. subtilis* DE111. There are no common outcomes that are consistent in all reported trials. Improvements in stool type were observed by Cuentas et al. [49]. Only one study reported a significant reduction in fat % in participants consuming probiotic [51], while the rest of the studies did not record any changes in anthropometric characteristics. Trotter et al. [53] observed significant improvements in some of the blood lipid biomarkers; however, three other studies that measured the same markers did not find any changes in lipid profile following supplementation. A significant increase in microbiota alpha-diversity was observed in children's stool samples post-supplementation [48] but not in the stools of the adult population [47]. However, these discrepancies can generally be explained by the heterogeneity of participants and differences in their baseline characteristics. Therefore, future studies should begin to look at individual characteristics that are drivers of response to probiotic intervention. Generally, the sample sizes of the studies reported here are relatively small; however, comparative analyses of comprehensive metadata across multiple studies can help identify common characteristics associated with positive responses to *B. subtilis* supplementation. Another approach, which requires larger sample sizes, is to select subsets with specific symptoms as a secondary analysis from the total tested population. In the present review, eligibility criteria often included participants who had "abdominal bloating, burping, or flatulence" [58] or "self-reported diagnosis of functional gastrointestinal disorders" [47,59], but how these were quantified and accounted for in screening is unclear. Subset analyses could determine if the magnitude of responses is associated with the severity of symptoms at baseline. Since some studies also identified improvements in lipid profiles, immune cell populations, reduction in proinflammatory biomarkers, etc., future research targeting populations with above-normal parameters (i.e., prediabetic, hypercholesterolemia, high baseline CRP, etc.) that are at high risk for chronic disease development is also warranted. An additional avenue for exploration is based on the specificity of effects of different strains of *B. subtilis* on gut microbiome. Microbiota modifications seem to be a major player in many GI tracts and other organ system disorders, including gastroparesis and endometriosis in conjunction with IBS, as highlighted in recent reviews [63,64]. The effects of *B. subtilis* as a transient probiotic but not a permanent GI resident is largely unexplored in the context of these diseases, which opens a wide area for future clinical research. Finally, it is worth noting that the studies reviewed here reinforce the safety and tolerability of probiotic *B. subtilis*. There is no indication that this probiotic poses additional risks beyond those of canonical probiotic species of *Bifidobacterium* or *Lactobacillus*. However, there are certain populations, such as people with open surgical wounds or severely ill or immunocompromised individuals for whom any probiotic consumption is contraindicated.

Despite inconsistencies in the reported benefits of *B. subtilis* strains, there was universal agreement that the probiotic is generally safe and tolerable for healthy humans, including children and older adults. This is important to note since the shelf-stable, heat- and chemical-resistant spores make *Bacillus subtilis* an ideal component of probiotic-supplemented foods. Indeed, it can already be found in numerous commercial food products and is likely to be incorporated in many future packaged food offerings. It would also be interesting to look at whether *B. subtilis* offers benefits outside of the GI tract. It is already a standard component of many biological amendments used in agriculture for improved plant growth and health. *Bacillus subtilis* can also be found in probiotics for pets and livestock. Future research could begin to explore benefits in other niches, such as on human skin or in the oral cavity.

## 5. Conclusions

Spore-based probiotic strains of *B. subtilis* are transient members of the human microbiota that cannot colonize the GI tract. However, they are well equipped to complete their entire life cycle within the environment of a human gut, exerting effects that can be attributed to either spores, vegetative cells, or both. Different strains of *B. subtilis* supplementation in human populations elicit an array of responses that can be classified as either positive or neutral (no change). Though some adverse events are listed in the results of the studies (that may be related or unrelated to treatment), no consistent negative effects were observed from *B. subtilis* consumption at any reported doses. This makes *B. subtilis* one of the most promising spore-based probiotic species for human applications. Future research should focus on identifying individual drivers of response/target subpopulations that would most likely experience benefits from consuming these probiotics. This approach will move probiotic supplementation closer to "precision nutrition" to enhance positive outcomes and health benefits.

**Author Contributions:** Conceptualization, N.W. and T.L.W.; writing—original draft preparation, N.W.; writing—review and editing, N.W. and T.L.W.; visualization, N.W.; supervision, T.L.W.; funding acquisition, T.L.W. All authors have read and agreed to the published version of the manuscript.

**Funding:** This research was funded by National Institutes of Health (NIH)-National Heart, Lung, Blood Institute (NHLBI) grant number 5R01HL144611.

**Conflicts of Interest:** T.L.W. is a member of the Scientific Advisory Board for Deerland Enzymes and Probiotics, a division of ADM.

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
