# Peer review of "Spore-Based Probiotic Bacillus subtilis: Current Applications in Humans and Future Perspectives"

_fermentation, doi:10.3390/fermentation10020078_

Round 1

Reviewer 1 Report

Comments and Suggestions for Authors

Fermentation Manuscript ID: fermentation-2814758

Type of manuscript: Review

Title: Spore-Based Probiotic Bacillus subtilis: Current Applications in Humans and Future Perspectives

Brief summary

Bacillus subtilis (B. subtilis) is an aerobic or facultative anaerobic Gram-positive rod-shaped bacterium disseminated ubiquitously in a wide type of environments whose life cycle includes a dormant phase in the form of endospore that can germinate to generate vegetative form metabolically active. After being administrated by oral route, B. subtilis spores germinate into vegetative stage and both forms have brief intestinal staying ant then they are cleared shortly after from the intestine. This narrative review highlighted the probiotic effects of spores from a single B. subtilis species formulated in capsules whether alone or containing symbiotic or post-biotics administered orally in healthy humans or in individuals with gastrointestinal dysfunctions. Evidence indicated the potential application of spore-based probiotic strains of B. subtilis to control intestinal diseases and to strength the intestinal homeostasis.

Highlights

A very informative review, very well written and of high pharmacologic and clinical impact in the development of products containing probiotics designed for the control of gastrointestinal diseases.

Minor comments

Line 164

Clarify please whether in most cases these assays were conducted in individuals orally treated with B. subtilis spores formulated in capsules

Line 174 (Table 1)

Number of references must be included instead of author name

Line 180

include and define please "HbA1c"  

Line 293

deleted please "(Trotter)"

Line 367

¿should it be "GITQ"?

Line 423

include please in brief, a presumable mechanism upon which amino acids as synbiotics enhance  the beneficial effects of spore-base probiotics derived from B. subtilis 

Line 456

define please "HbA1c"

Line 526

It is advisable to state the human population which the spore-base probiotic of B- subtilis migh provide potential risks so that it administration would not be recommendable

Reviewer 2 Report

Comments and Suggestions for Authors

Thank you for inviting me to review this paper, which describes B. subtilis in the context of its use as a probiotic for improving human gastrointestinal health. The review is well-written, the English is very fluent, and the organization of the paragraphs is absolutely correct.

From my point of view, the only modification I request is a comment throughout the text, a clinical clarification of the gastrointestinal symptoms or disease which benefit from Bacillus administration, which is not well understood from the text. It would be important, according to this modification, to clarify the influence of Bacillus on functional GI diseases, such as gastroparesis and IBS. In the context of the hypothesis that Bacillus acts by modifying the microbiota, the authors might consider including the following references in the text: PMID 37630649 and 37317096. It would be interesting in future conclusions to hypothesize the role of Bacillus in the treatment of functional gastrointestinal conditions.

Comments on the Quality of English Language

Minor editing of English is required. 

Author Response

Please see attachment for reviewer responses.
